# Private Synthetic Data for Multitask Learning and Marginal Queries

**Giuseppe Vietri** *
University of Minnesota
vietr002@umn.edu

**Cedric Archambeau**
Amazon AWS AI/ML
cedrica@amazon.com

**Sergul Aydore**
Amazon AWS AI/ML
saydore@amazon.com

**William Brown**[†]
Columbia University
w.brown@columbia.edu

**Michael Kearns**
Amazon AWS AI/ML
University of Pennsylvania
mkearns@cis.upenn.edu

**Aaron Roth**
Amazon AWS AI/ML
University of Pennsylvania
aaroth@cis.upenn.edu

**Ankit Siva**
Amazon AWS AI/ML
ankitsiv@amazon.com

**Shuai Tang**
Amazon AWS AI/ML
shuat@amazon.com

**Zhiwei Steven Wu**
Amazon AWS AI/ML
Carnegie Mellon University
zhiweiw@andrew.cmu.edu

## Abstract

We provide a differentially private algorithm for producing synthetic data simultaneously useful for multiple tasks: marginal queries and multitask machine learning (ML). A key innovation in our algorithm is the ability to directly handle numerical features, in contrast to a number of related prior approaches which require numerical features to be first converted into high cardinality categorical features via a binning strategy. Higher binning granularity is required for better accuracy, but this negatively impacts scalability. Eliminating the need for binning allows us to produce synthetic data preserving large numbers of statistical queries such as marginals on numerical features, and class conditional linear threshold queries. Preserving the latter means that the fraction of points of each class label above a particular half-space is roughly the same in both the real and synthetic data. This is the property that is needed to train a linear classifier in a multitask setting. Our algorithm also allows us to produce high quality synthetic data for mixed marginal queries, that combine both categorical and numerical features. Our method consistently runs 2-5x faster than the best comparable techniques, and provides significant accuracy improvements in both marginal queries and linear prediction tasks for mixed-type datasets.

## 1 Introduction

We study the problem of synthetic data generation subject to the formal requirement of differential privacy [DMNS06]. Private synthetic data have the advantage that they can be reused without any further privacy cost. As a result, they can use a limited privacy budget to simultaneously enable a wide variety of downstream machine learning and query release tasks.

---

[*]Giuseppe is the lead author; all other authors are listed in alphabetical order. Giuseppe performed this work during an internship at AWS AI/ML

[†]William performed this work during an internship at AWS AI/ML.

36th Conference on Neural Information Processing Systems (NeurIPS 2022).

Recent work on practical methods for private synthetic data has largely split into two categories. The first category builds on the empirical success of deep generative models and develops the corresponding private implementations, such as DP-GAN [XLW+18, BJWW+19, NWD20, JYvdS18]. The second category follows the principle of *moment-matching* and generates synthetic data that preserve a large family of marginal queries [HLM12, GAH+14, MSM19, MPSM21, ABK+21, MMSM22, LVW21]. For both downstream machine learning and query release tasks, the moment-matching approach has generally outperformed private deep generative models [TMH+21], which are known to be difficult to optimize subject to differential privacy [NWD20]. In order to handle numerical features, they are discretized into a finite number of categorical bins. This binning heuristic is impractical for real-world datasets, which often contain a large number numerical features with wide ranges.

Our work develops a scalable, differentially private synthetic data algorithm, called RAP++, that builds on the previous framework of *"Relaxed Adaptive Projection"* (RAP) [ABK+21] but can handle a mixture of categorical and numerical features without any discretization. As with RAP, the core algorithmic task of RAP++ is to use differentiable optimization methods (e.g. SGD or Adam) to find synthetic data that most closely matches a noisy set of query measurements on the original data. We introduce new techniques to be able to do this for queries defined over a mixture of categorical and numerical features. Two key components that drive the success of our approach includes (+) random linear projection queries to handle mixed-type data, and (+) tempered sigmoid annealing on top of the existing work RAP. Hence, we refer to our algorithm as RAP++. Our contributions can be summarized as follows:

**Mixed-type queries.** To capture the statistical relationships of a mixture of categorical and numerical features, we consider two classes of statistical queries on continuous data that previous work can only handle through binning of real values into high cardinality categorical features. The first class of queries we consider is *mixed marginals*, which capture the marginal distributions over subsets of features. Second, we define the class of *class-conditional linear threshold functions*, which captures how accurate any linear classifier is for predicting a target label (which can be any one of the categorical features). Therefore, we can generate synthetic data for multitask learning by choosing a suitable set of class-conditioned queries, where the conditioning is on multiple label columns of the data. The inability of previous work to handle mixed-type data without binning has been acknowledged as an important problem [MMSM22], and our approach takes a significant step towards solving it.

**Tempered sigmoid annealing.** Since both classes of mixed-type queries involve threshold functions that are not differentiable, we introduce sigmoid approximation in order to apply the differentiable optimization technique in RAP. The sigmoid approximation is a smooth approximation to a threshold function of the form $\frac{1}{1+\exp(-\sigma(x-b))}$, where $\sigma$ is called the inverse temperature parameter controlling the slope of the function near the threshold $b$. However, choosing the right parameter $\sigma$ involves a delicate trade-off between approximation and optimization. To balance such trade-offs, we provide an adaptive gradient-based optimization method that dynamically increases the inverse temperature $\sigma$ based on the gradient norms of iterates. We show that our method can still reliably optimize the moment-matching error even when no fixed value of $\sigma$ can ensure both faithful approximation and non-vanishing gradient.

**Empirical evaluations.** We provide comprehensive empirical evaluations comparing RAP++ against several benchmarks, including PGM [MSM19], DP-CTGAN [FDK22], and DP-MERF [HAP21] on datasets derived from the US Census. In terms of accuracy, RAP++ outperforms these benchmark methods in preserving the answers for the two classes of mixed-type queries. We also train linear models for multiple classification tasks using the synthetic datasets generated from different algorithms. We find that RAP++ provides the highest accuracy when the numeric features are predictive of the target label, and closely tracks all benchmark accuracy in all other cases.

## 2 Preliminaries

We use $\mathcal{X} = \mathcal{X}_1 \times \ldots \times \mathcal{X}_d$ to denote a $d$-dimensional data domain, where $i$-th feature has domain $\mathcal{X}_i$. For a set $S$ of features, we denote the projected domain onto $S$ as $\mathcal{X}_S = \prod_{i \in S} \mathcal{X}_i$. This work assumes that each feature domain $\mathcal{X}_i$ can be either categorical or continuous. If a feature $i$ is categorical, it follows that $\mathcal{X}_i$ is a finite unordered set with cardinality $|\mathcal{X}_i|$ and if it is continuous then $\mathcal{X}_i = \mathbb{R}$. A dataset $D \in \mathcal{X}^*$ is a multiset of points of arbitrary size from the domain. For any point $x \in \mathcal{X}$, we

use $x_i \in \mathcal{X}_i$ to denote the value of the $i$-th feature and for any set of features $S$, let $x_S = (x_i)_{i \in S}$ be the projection of $x \in \mathcal{X}$ onto $\mathcal{X}_S$. The algorithm in this work is based on privately releasing *statistical queries*, which are formally defined here.

**Definition 1** (Statistical Queries [Kea98]). *A statistical query (also known as a linear query or counting query) is defined by a function $q : \mathcal{X} \to [0, 1]$. Given a dataset $D$, we denote the average value of $q$ on $D$ as: $q(D) = \frac{1}{|D|} \sum_{x \in D} q(x)$.*

## 2.1 Mixed-type Data and Threshold Queries

In this section we consider multiple classes of statistical queries of interest. Most previous work on private synthetic data generation focuses on categorical data and in particular preserving marginal queries, which are formally defined as follows:

**Definition 2** (Categorical Marginal queries ). *A $k$-way marginal query is defined by a set of categorical features $S$ of cardinality $|S| = k$, together with a particular element $c \in \prod_{i \in S} \mathcal{X}_i$ in the domain of $S$. Given such a pair $(S, c)$, let $\mathcal{X}(S, c) = \{x \in \mathcal{X} : x_S = c\}$ denote the set of points that match $c$. The corresponding statistical query $q_{S,c}$ is defined as $q_{S,c}(x) = \mathbb{1}\{x \in \mathcal{X}(S, c)\}$, where $\mathbb{1}$ is the indicator function.*

Prior work on private query release [BLR08, GAH⁺14, MSM19, ABK⁺21, MMSM22, VTB⁺20, GAH⁺14, LVW21] considered only categorical marginal queries and handled mixed-type data by binning numerical features. This work considers query classes that model relations between categorical and numerical features, which we call *Mixed Marginals*, without needing to discretize the data first. One example of a *Mixed Marginal* query is: *"the number of people with college degrees and income at most \$50K"*. This example is mixed-marginal because it references a categorical feature (i.e., education) and a numerical feature (i.e., income).

**Definition 3** (Mixed Marginal Queries). *A $k$-way mixed-marginal query is defined by a set of categorical features $C$, an element $y \in \mathcal{X}_C$, a set of numerical features $R$ and a set of thresholds $\tau$, with $|C| + |R| = k$ and $|R| = |\tau|$. Let $\mathcal{X}(C, y)$ be as in definition 2 and let $\mathcal{X}(R, \tau) = \{x \in \mathcal{X} : x_j \leq \tau_j \quad \forall_{j \in R}\}$ denote the set of points where each feature $j \in R$ fall below its corresponding threshold value $\tau_j$. Then the statistical query $q_{C,y,R,\tau}$ is defined as*

$$q_{C,y,R,\tau}(x) = \mathbb{1}\{x \in \mathcal{X}(C, y)\} \cdot \mathbb{1}\{x \in \mathcal{X}(R, \tau)\}.$$

Another natural query class that we can define on continuous valued data is a linear threshold query — which counts the number of points that lie above a halfspace defined over the numeric features. A *class conditional* linear threshold query is defined by a target feature $i$, and counts the number of points that lie above a halfspace that take a particular value of feature $i$.

**Definition 4** (Class Conditional Linear Threshold Query). *A class conditional linear threshold query is defined by a categorical feature $i$, a target value $y \in \mathcal{X}_i$, a set of numerical features $R$, a vector $\theta \in \mathbb{R}^{|R|}$ and a threshold $\tau \in \mathbb{R}$ as $q_{i,y,R,\theta,\tau}(x) = \mathbb{1}\{\langle \theta, x_R \rangle \leq \tau \text{ and } x_i = y\}$.*

The quality of synthetic data can be evaluated in task-specific ways. Since the goal is to accurately answer a set of queries over numerical features, we can evaluate the difference between answers to the queries on the synthetic data and those on the real data, summarized by an $\ell_1$ norm.

**Definition 5** (Query Error). *Given a set of $m$ statistical queries $Q = \{q_1, \ldots, q_m\}$, the average error of a synthetic dataset $\widehat{D}$ is given by: $\frac{1}{m} \sum_{i=1}^{m} |q_i(D) - q_i(\widehat{D})|$.*

## 2.2 Differential Privacy

The notion of privacy that we adopt in this paper is differential privacy, which measures the effect of small changes in a dataset on a randomized algorithm. Formally, we say that two datasets are neighboring if they are different in at most one data point.

**Definition 6** (Differential Privacy [DMNS06]). *A randomized algorithm $\mathcal{M} : \mathcal{X}^n \to \mathcal{R}$ satisfies $(\epsilon, \delta)$-differential privacy if for all neighboring datasets $D, D'$ and for all outcomes $S \subseteq \mathcal{R}$ we have*

$$Pr\left[\mathcal{M}(D) \in S\right] \leq e^{\epsilon} Pr\left[\mathcal{M}(D') \in S\right] + \delta.$$

In our analysis we adopt a variant of DP known as (zero) Concentrated Differential Privacy which more tightly tracks composition and can be used to bound the differential privacy parameters $\epsilon$ and $\delta$:

**Definition 7** (Zero Concentrated Differential Privacy [BS16a]). *A randomized algorithm* $\mathcal{M} : \mathcal{X}^n \to \mathcal{R}$ *satisfies $\rho$-zero Concentrated Differential Privacy ($\rho$-zCDP) if for all neighboring datasets $D, D'$, and for all $\alpha \in (0, \infty)$: $\mathbb{D}_\alpha(\mathcal{M}(D), \mathcal{M}(D')) \leq \rho\alpha$, where $\mathbb{D}_\alpha(\mathcal{M}(D), \mathcal{M}(D'))$ is $\alpha$-Renyi divergence between the distributions $\mathcal{M}(D)$ and $\mathcal{M}(D')$.*

We use two basic DP mechanisms that provide the basic functionality of selecting high information queries and estimating their answers. To answer statistical queries privately, we use the Gaussian mechanism, which we define in the context of statistical queries:

**Definition 8** (Gaussian Mechanism). *The Gaussian mechanism $\mathcal{G}(D, q, \rho)$ takes as input a dataset $D \in \mathcal{X}^*$, a statistical query $q : \mathcal{X}^* \to [0, 1]$, and a zCDP parameter $\rho$. It outputs noisy answer $\hat{a} = q(D) + Z$, where $Z \sim \mathcal{N}(0, \frac{1}{2n^2\rho})$, where $n$ is the number of rows in $D$.*

**Lemma 1** (Gaussian Mechanism Privacy [BS16a]). *For any statistical query $q$, and parameter $\rho > 0$, the Gaussian mechanism $\mathcal{G}(\cdot, q, \rho)$ satisfies $\rho$-zCDP.*

Answering all possible queries from a large set may be expensive in terms of privacy, as the noise added to each query scales polynomially with the number of queries. A useful technique from [GHRU11] is to iteratively construct synthetic data by repeatedly selecting queries on which the synthetic data currently represents poorly, answering those queries with the Gaussian mechanism, and then re-constituting the synthetic data. Thus, we need a private selection mechanism. We use the *Report Noisy Top-$K$* mechanism [DR19], defined here in context of selecting statistical queries.

**Definition 9** (One-shot Report Noisy Top-$K$ With Gumbel Noise). *The "Report Noisy Top-$K$" mechanism $\mathrm{RN}_K(D, \widehat{D}, Q, \rho)$, takes as input a dataset $D \in \mathcal{X}^n$ with $n$ rows, a "synthetic dataset" $\widehat{D} \in \mathcal{X}^*$, a set of $m$ statistical queries $Q = \{q_1, \ldots, q_m\}$, and a zCDP parameter $\rho$. First, it adds Gumbel noise to the error of each $q_i \in Q$:*

$$\hat{y}_i = \left| q_i(D) - q_i(\widehat{D}) \right| + Z_i, \text{ where } \quad Z_i \sim Gumbel\left(K/\sqrt{2\rho}n\right),$$

*Let $i_{(1)}, \ldots, i_{(m)}$ be an ordered set of indices such that $\hat{y}_{i_{(1)}} \geq, \ldots, \geq \hat{y}_{i_{(m)}}$. The algorithm outputs the top-$K$ indices $\{i_{(1)}, \ldots, i_{(K)}\}$ corresponding to the $K$ queries where the answers between $D$ and $\hat{D}$ differ most.*

**Lemma 2** (Report Noisy Top-$K$ Privacy [ABK+21]). *For a dataset $D$, a synthetic dataset $\widehat{D}$, a set of statistical queries $Q$, and zCDP parameter $\rho$, $\mathrm{RN}_K(D, \widehat{D}, Q, \rho)$ satisfies $\rho$-zCDP.*

## 3 Relaxed Projection with Threshold Queries

In this section we propose a gradient-based optimization routine for learning a mixed-type synthetic datasets that approximate answers to threshold based queries over continuous data. The technique is an extension of the *relaxed projection* mechanism from [ABK+21], which in turn extends the *projection mechanism* of [NTZ13].

Given a dataset $D$, a collection of $m$ statistical queries $Q = \{q_1, \ldots, q_m\}$ and zCDP parameter $\rho$, the projection mechanism [NTZ13] consists of two steps: (1) For each query index $i \in [m]$, evaluate $q_i$ on $D$ using the Gaussian mechanism: $\hat{a}_i = G(D, q_i, \rho/m)$, and then (2) Find a synthetic dataset $D' \in \mathcal{X}^*$ whose query values minimize the distance to the noisy answers $\{\hat{a}_i\}_{i \in [m]}$:

$$\arg \min_{D' \in \mathcal{X}^*} \sum_{i \in [m]} \left(\hat{a}_i - q_i(D')\right)^2. \tag{1}$$

The projection step produces synthetic data that implicitly encodes the answers to all queries in the query set $Q$. In addition to producing synthetic data (which can be used for downstream tasks like machine learning which cannot easily be accomplished with the raw outputs of the Gaussian mechanism), by producing a synthetic dataset, the projection step by definition enforces consistency constraints across all query answers, which is accuracy improving. Unfortunately, the projection step is generally an intractable computation since it is a minimization of a non-convex and non-differentiable objective over an exponentially large discrete space. To address this problem, [ABK+21] gives an algorithm that relaxes the space of datasets $\mathcal{X}^n$ to a continuous space, and generalizes the statistical queries to be differentiable over this space:

**Domain relaxation.** The first step is to represent categorical features as binary features using one-hot encoding. Let $d' := \sum_{i \in C} |\mathcal{X}_i|$, where $C$ is the set of categorical features, be the dimension of the

feature vector under the one-hot encoding. Then, we consider a continuous relaxation denoted by $\widetilde{\mathcal{X}}$ of the one-hot encoding feature space. Using $\Delta(\mathcal{X}_i)$ to denote a probability space over the elements of $\mathcal{X}_i$, we choose $\widetilde{\mathcal{X}} := \prod_{i \in C} \Delta(\mathcal{X}_i)$ to be the product measure over probability distributions over the one-hot encodings. The advantage of representing the relaxed domain as a probability space is that one can quickly recover a dataset in the original discrete domain by random sampling.

**Differentiable queries.** The next step is to replace the set of queries $Q$ by a set of continuous and differentiable queries $\widetilde{Q}$ over the relaxed domain $\widetilde{\mathcal{X}}$. Informally, in order to solve our optimization problem, the queries must be differentiable over the relaxed domain $\tilde{\mathcal{X}}$ (and have informative gradients), but must also be a good approximation to the original queries in $Q$ so that optimizing for the values of the relaxed queries produces a synthetic dataset that is representative of the original queries. For categorical marginal queries following [ABK$^+$21], we use the set of product queries over the relaxed domain which are equal to categorical marginal queries on the original domain:

**Definition 10.** *Given a subset of features $T \subseteq [d']$ of the relaxed domain $\widetilde{\mathcal{X}}$, the product query $q_T : \widetilde{\mathcal{X}}$ is defined as $q_T(x) = \prod_{i \in T} x_i$.*

Next we give a differentiable relaxation for any class of threshold based statistical queries such as those in definition 3 and definition 4. A linear threshold query is a threshold applied to a linear function of the data, which is not differentiable. We choose to approximate thresholds with sigmoid functions. They are a simple parametric class of functions with well-behaved gradients with adjustable magnitudes for the approximation error via the *inverse temperature* parameter:

**Definition 11** (Tempered Sigmoid)**.** *The sigmoid threshold function $f_\tau^{[\sigma]} : \mathbb{R} \to [0, 1]$ with threshold $\tau \in \mathbb{R}$ and inverse temperature $\sigma > 0$ is defined as $f_\tau^{[\sigma]}(x) = \left( \frac{1}{1 + e^{-\sigma(x-\tau)}} \right)$ for any $x \in \mathbb{R}$.*

The sigmoid function in definition 11 is a differentiable approximation of the class of 1-way prefix marginal queries, which serves as a basic building block for approximating other interesting classes of threshold-based queries, such as the class of *conditional linear threshold* queries.

**Definition 12** (Tempered Sigmoid Class Conditional Linear Threshold Queries)**.** *A sigmoid conditional linear threshold query is defined by a categorical feature $i \in [d']$ of the relaxed domain space, a set of numerical features $T \subseteq [d']$ in the relaxed domain space, a vector $\theta \in \mathbb{R}^{|T|}$, and threshold $\tau \in \mathbb{R}$. Fixing the sigmoid temperature $\sigma$, let $f_\tau^{[\sigma]}$ be defined as in definition 11 then the corresponding differentiable statistical query is $q_{i,T,\theta,\tau}^{[\sigma]}(x) = x_i \cdot f_\tau^{[\sigma]}(\langle \theta, x_T \rangle)$.*

By relaxing the data domain to its continuous representation $\widetilde{\mathcal{X}}^*$ and replacing the query set $Q = \{q_1, \ldots, q_m\}$ by its differentiable counterpart $\widetilde{Q} = \{\tilde{q}_1, \ldots, \tilde{q}_m\}$ we obtain the new objective:

$$\arg\min_{D' \in \widetilde{\mathcal{X}}^*} \widetilde{L}(D') := \sum_{i \in [m]} (\hat{a}_i - \tilde{q}_i(D'))^2 \tag{2}$$

Since the new objective is differentiable everywhere in $\widetilde{\mathcal{X}}^*$, we can run any first-order continuous optimization method to attempt to solve (2). Were $\tilde{q}_i$ a good approximation for $q_i$ *everywhere* in $\widetilde{\mathcal{X}}$ for all queries $i \in [m]$, then the solution to eq. (2) would approximate the solution to eq. (1). However, there are two other challenges that arise when we use the sigmoid approximation: flat gradients and near-threshold approximation quality. If the sigmoid function has a large $\sigma$ (inverse temperature), its derivatives at points far from the threshold $\tau$ have small magnitudes. In other words, sigmoid approximations to linear threshold functions have nearly flat gradients far from the threshold. In the flat gradient regime, first order optimization algorithms fail because they get "stuck". This can be mitigated by using a small $\sigma$. But there is another issue: because threshold functions are discontinuous, any continuous approximation to a threshold function must be

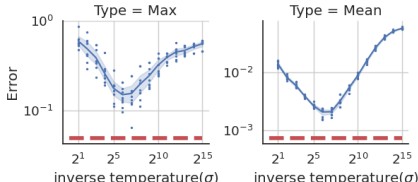

Figure 1: Comparison of the Maximum error (left) and the Mean error (right) of optimizing mixed marginals using our temperature annealing (red line) and using a fixed $\sigma$ parameter (blue curve). It is clear that annealing the $\sigma$ parameter strategically during optimization leads to lower error than using any fixed $\sigma$ parameter during the continuous projection step. The error is over a set of 1000 random 2-way mixed-marginal queries (see definition 3). And the underlying dataset is described in detail in section 5.

---

**Algorithm 1** Relaxed Projection with Sigmoid Temperature Annealing

---

1: **Input:** A set of $m$ sigmoid differentiable queries $\{\tilde{q}_i^{[\cdot]}\}_{i\in[m]}$, a set of $m$ target answers $\hat{a} = \{\hat{a}_i\}_{i\in[m]}$, initial inverse temperature $\sigma_1 \in \mathbb{R}^+$, stopping condition $\gamma > 0$, and initial dataset $\widehat{D}_1$.
2: **for** $j = 1$ **to** $J$ **do**
3:     Set inverse temperature $\sigma_j = \sigma_1 \cdot 2^{j-1}$.
4:     Define the sigmoid differentiable loss function: $\mathrm{L}_j(\widehat{D}) = \sum_{i\in[m]} \left( \tilde{q}_i^{[\sigma_j]}(\widehat{D}) - \hat{a}_i \right)^2$
5:     Starting with $\widehat{D} \leftarrow \widehat{D}_j$. Run gradient descent on $L_j(\widehat{D})$ until $\|\nabla \mathrm{L}_j(\widehat{D})\| \le \gamma$. Set $\widehat{D}_{j+1} \leftarrow \widehat{D}$.
6: **end for**
7: **Output** $\widehat{D}_{J+1}$

---

**Algorithm 2** Relaxed Adaptive Projection + Mixed Type + Temperature Annealing (RAP++ )

---

1: **Input** A dataset $D$ with $n$ records, a collection of $m$ statistical queries $Q = \{q_1, \ldots, q_m\}$, query samples per round $K \le m$, number of adaptive epochs $T \le m/K$, the size of the synthetic dataset $\hat{n}$, a sigmoid temperature parameter $\sigma_1$ and differential privacy parameters $\varepsilon, \delta$.
2: Let $\rho$ by such that: $\varepsilon = \rho + 2\sqrt{\rho \log(1/\delta)}$.
3: Initialize relaxed dataset $\widehat{D}_1 \in \widetilde{\mathcal{X}}^{\hat{n}}$ uniformly at random, and $\sigma_1 \in \mathbb{R}^m$.
4: **for** $t = 1$ **to** $T$ **do**
5:     Choose $K$ queries $\{q_{t,j}\}_{j\in[K]} \subset Q$ using $\mathrm{RN}_K(D, \widehat{D}_t, Q \setminus Q_{t-1}, \frac{\rho}{2T})$.
6:     For each $j \in [K]$, $\hat{a}_{t,j} \leftarrow \mathcal{G}\left(D, q_{t,j}, \left(\frac{\rho}{2TK}\right)\right)$.
7:     Let $Q_t = \{q_{i,j}\}_{i\in[t],j\in[K]}$ and $\hat{a}_t = \{\hat{a}_{i,j}\}_{i\in[t],j\in[K]}$.
8:     Let $\widetilde{Q}_t$ be the set of differentiable queries corresponding to $Q_t$
9:     Project step: $\widehat{D}_{t+1} \leftarrow$ RP-Sigmoid-Temperature-Annealing$(\widetilde{Q}_t, \hat{a}_t, \widehat{D}_t, \sigma_1, \gamma)$.
10: **end for**
11: **Output:** $\widehat{D}_{T+1}$

---

a poor approximation for points that are sufficiently close to the threshold. To make this poor approximation regime arbitrarily narrow, we would have to choose a large $\sigma$, but this is in tension with the flat gradients problem mentioned earlier. As a result, it is not clear how to choose an optimal value for $\sigma$.

We overcome this issue by using a "temperature annealing" approach, described in detail in algorithm 1. Informally, we start with a small $\sigma$, and run our optimization until the magnitude of the gradients of our objective function fall below a pre-defined threshold. At that point, we double $\sigma$ and repeat, until convergence. The intuition behind this technique is that if the magnitude of the gradients has become small, we must be near a local optimum of the relaxed objective (eq. (2)), since we have stopped making progress towards the relaxed objective for the current value of $\sigma$. However, this might not be close to a local optimum of the actual objective function eq. (1). By increasing $\sigma$ at this stage, we make the relaxed objective a closer approximation to the real objective; we continue optimization until we are close to a local optimum of the new relaxed objective, and then repeat. We find that this annealing approach worked well in practice. Figure 1 shows comparison of temperature annealing to optimization with fixed sigmoid temperature parameters. It can be seen that the annealing approach improves over every fixed setting of the parameter.

## 4 Threshold Query Answering with RAP++

We described the tools for optimizing the relaxed objective in Eq. 2. In this section, we introduce our private synthetic data generation algorithm RAP++ for mixed-type data (Alg. 2). RAP++ accepts as input mixed-type data and supports releasing answers to both mixed marginal and class conditional threshold queries. Our approach for synthetic data generation differs from prior work in that previous mechanisms [MSM19, MMSM22, GAH+14, VTB+20, ABK+21] only operate on discrete data for answering categorical marginal queries. For mixed-type data, one could simply discretize the data and run one of the previously known mechanisms, however, we will show that directly optimizing over threshold based queries has an advantage in terms of accuracy and scalability.

Given an input dataset $D \in \mathcal{X}^*$ and a collection of statistical queries $Q$, RAP++ operates over a sequence of $T$ rounds to produce a synthetic dataset that approximates $D$ on the queries $Q$. First,

RAP initializes a relaxed synthetic dataset $\widehat{D}_1 \in \mathcal{X}^{\hat{n}}$, of size $\hat{n}$, uniformly at random from the data domain of the original dataset. On each round $t = 1, \ldots, T$, the algorithm then calls the RN mechanism to select $K$ queries (denoted by $Q_t$) on which the current synthetic data $\widehat{D}_t$ is a poor approximation, and uses the Gaussian mechanism to privately estimate the selected queries. Using these new queries and noisy estimates, the algorithm calls algorithm 1 to solve (2) and find the next synthetic dataset $\widehat{D}_{t+1}$, such that $\widehat{D}_{t+1}$ is consistent with $D$ on the current set of queries $Q_t$. The process continues until the number of iterations reach $T$.

The way algorithm RAP is described in algorithm 2 is specific to the setting of answering threshold-based queries, since it uses a temperature annealing step which applies only to queries which use sigmoid threshold functions like mixed marginals (definition 3) or linear threshold queries (definition 4). We can also use the algorithm to handle non-threshold based queries such as categorical marginals — but in this case our algorithm reduces to the RAP algorithm due to [ABK+21]. We can also run the algorithm using multiple query classes, using threshold annealing for those query classes on which it applies.

**Differential Privacy**   The algorithm's privacy analysis follows from composition of a sequence of RN and Gaussian mechanisms, which is similar to other approaches that select queries adaptively [GHRU11, GRU12, HLM12, ABK+21, VTB+20, LVW21]. The following theorem states the privacy guarantee.

**Theorem 1** (Privacy analysis of RAP++). *For any dataset $D$, any query class $Q$, any set of parameters $K$, $T$, $\hat{n}$, $\sigma_1$, and any privacy parameters $\epsilon, \delta > 0$, Algorithm 2 satisfies $(\epsilon, \delta)$-differential privacy.*

Proof of theorem 1 follows the composition property of $\rho$-zCDP. We defer the proof to the appendix.

**Accuracy**   The accuracy of our method (algorithm 2) for answering a collection of statistical queries depends mainly on the success of our optimization oracle (algorithm 1). Since the oracle algorithm 1 is a heuristic, we cannot provide a formal accuracy guarantee. However, the paper provides empirical evidence of the oracle's performance.

We remark that the work of [ABK+21] provided accuracy guarantees for RAP under the assumption that their oracle solves the optimization step perfectly. Our method, which is an instantiation of RAP with new query classes, inherits RAP 's accuracy guarantees (again, under the assumption that the optimization problem (1) can be solved).

## 5   Experiments

The experiments are performed over a collection of mixed-type real-world public datasets. The quality of generated synthetic datasets is evaluated in terms of the error on a set of mixed-marginal queries as well as on their *usefulness* for training linear models using logistic regression. We compare our method RAP++ against existing well-known algorithms for synthetic data generation, including PGM[MSM19], DP-MERF [HAP21], CTGAN [FDK22], RAP [ABK+21]. We use the adaptive version of PGM, which is called MWEM+PGM in the original paper. We compare our algorithm RAP++ with all other algorithms at various privacy levels quantified by $\epsilon$, with $\delta$ always set to be $1/n^2$ for all algorithms.

**Datasets.** We use a suite of new datasets derived from US Census, which are introduced in [DHMS21]. These datasets include five pre-defined prediction tasks, predicting labels related to income, employment, health, transportation, and housing. Each task defines feature columns (categorical and numerical) and target columns, where feature columns are used to train a model to predict the target column. Each task consists of a subset of columns from the American Community Survey (ACS) corpus and the target column for prediction. We use the five largest states (California, New York, Texas, Florida, and Pennsylvania) which together with the five tasks constitute 25 datasets. We used the folktables package [DHMS21] to extract features and tasks.[3] In the appendix, we include a table that summarizes the number of categorical and numerical features for each ACS task and the number of rows in each of the 25 datasets.

---

[3]The Folktables package comes with MIT license, and terms of service of the ACS data can be found here: https://www.census.gov/data/developers/about/terms-of-service.html.

We are interested in datasets that can be used for *multiple* classification tasks simultaneously. To this end, we create multitask datasets by combining all five prediction tasks. The result is five multitask datasets corresponding to five states, where each multitask dataset has five target columns for prediction. The learning problem given a multitask dataset per state consists of learning five different models, one to predict each of the target columns from the feature columns. Each multitask dataset has 25 categorical features, 9 numerical features, and five target binary labels.

## 5.1 Statistical Queries

Here, we describe three different classes of statistical queries that were used either for synthesizing datasets or for evaluation. First, we introduce some notations. Given as input a dataset with columns $[d]$, let $C, R, L$ be a column partition of $[d]$, where $C$, $R$, and $L$ denote the categorical features, numerical features, and target columns for ML tasks, respectively.

**Class Conditional Categorical Marginals (CM).** This class of queries involve marginals defined over two categorical feature columns and one target column, making it a subset of 3-way categorical marginal queries. Using notation from definition 2, this query class is defined as $Q_{cat} = \{q_{S,y} : S \in (C \times C \times L), y \in \mathcal{X}_S\}$. This query class is constructed to preserve the relationship between feature columns and target columns with the goal of generating synthetic data useful for training ML models. Since the possible set of queries for this class is finite we enumerate over all possible combinations of 3-way marginals in our experiments.

**Class Conditional Mixed-Marginals (MM).** Similar to the class conditional categorical marginals, we create 3-way marginal queries involving two feature columns and one target column. For this query class, however, we use only numerical columns for the features as opposed to CMs. Since the possible set of marginals involving numerical features is infinite, we use $200,000$ random 3-way mixed marginal queries in our experiments.

**Class Conditional Linear Thresholds (LT).** Next we describe the construction of the *class conditional linear threshold queries* using definition 4. We generate a set $Q_{lin}$ of $m = 200,000$ random queries, which is populated as follows:

1. Generate a random vector $\hat{\theta} \in \mathbb{R}^{d_{|R|}}$, where the value of each coordinate $i \in R$ is sampled as $\hat{\theta}_i \sim \mathcal{N}(0,1)$. Then set $\theta = \hat{\theta}/\sqrt{d}$.
2. Sample a threshold value $\tau$ from the standard normal distribution, i.e., $\tau \sim \mathcal{N}(0,1)$.
3. Sample a label $i \in L$ and a target value for the label $y \in \mathcal{X}_i$.
4. Add the query $q_{i,y,R,\theta,\tau}$ to the set $Q_{lin}$.

In the experiments that follow, RAP++ is trained both with the CM queries and the *class conditional linear threshold* queries. PGM and RAP are trained with the CM queries over their discretized domain (i.e. we bin the numerical features so that they become categorical, then run the algorithms to preserve CM queries). We use MM queries mainly for evaluation (see fig. 2).

**Experimental Setup.** For both RAP and PGM we discretized all numerical features using an equal-sized binning, and compare the performance for numbers of bins in $\{10, 20, 30, 50, 100\}$. For the results, we choose 30 bins for discretization, as it performs well overall across different tasks and privacy levels. In the appendix we show more results for different choices of bin size. The other relevant parameter for both PGM and RAP is the number of epochs of adaptivity, which is fixed to be $d - 1$, where $d$ is the number of columns in the data.

Next we describe the relevant parameters used to train RAP++ . Since RAP++ optimizes over two query classes (CM and LT), the first parameter $T_{CM}$ corresponds to the number of adaptive epochs for selecting CM queries. The other parameter $T_{LT}$ corresponds to the number of adaptive epochs for selecting LT queries. To be consistent with PGM and RAP we always choose $T_{CM} = d_{|C|} - 1$, where $d_{|C|}$ is the number of categorical columns in the data. Finally, $K$ is the number of queries selected per epoch as described in algorithm 2.

We fix the parameters of RAP++ to be $T_{LT} = 50$ and $K = 10$ since it works well across all settings in our experiments. Figure 3 shows the effect of different hyperparameter choices for both RAP++ and PGM . It can be seen that RAP++ achieves better accuracy in less runtime than PGM . Furthermore, RAP++ is not very sensitive to varying $T_{LT}$, whereas PGM 's performance is very sensitive to the bin-size. For implementation details, see the appendix.

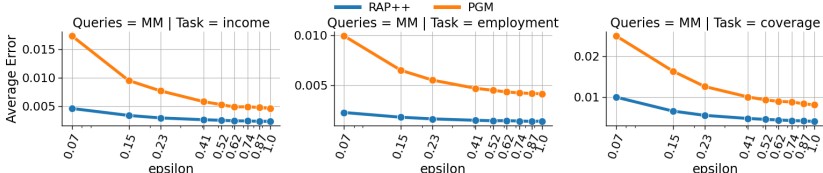

Figure 2: **Summary of average error** for Mixed Marginals (MM) on three tasks and aggregated over five ACS states. Each column represents different task. We only compare against the most competitive approach PGM . See appendix for other approaches and CM queries. RAP++ consistently achieves lower error, especially at small $\epsilon$ values.

## 5.2 Main Results

**Mixed-Marginals Evaluation.** We begin by comparing RAP++ against PGM on how well the generated synthetic data approximates a set of random MM queries on 25 single-task ACS datasets (see appendix). We use PGM as our primary baseline because it was the most performant of the various approaches we tried (DP-MERF , CTGAN , and RAP ). Here we show results on three tasks only for MM queries in fig. 2 and the remaining tasks, and CM query results are shown in the appendix. Note that in these experiments, the difference between RAP and RAP++ is in how it treats numeric valued features and threshold-based queries. In particular, for CM queries, RAP and RAP++ are essentially the same, so it does not make sense to compare RAP and RAP++ in this setting.

The figure shows average errors across all states for each task with respect to $\epsilon$. For each task/state/epsilon setting, we compute the average error over the set of queries (either CM or MM queries). Then we take another average over states for each task/epsilon. On average, RAP++ performed slightly better at answering CM queries, and significantly improved the average error on MM queries across all tasks. Note that none of the approaches used MM queries in training, hence the results indicate better generalization capability of RAP++ .

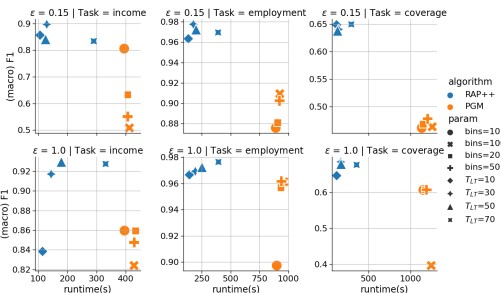

Figure 3: **F1 score vs. Runtime .** Comparing RAP++ (blue) and PGM (orange) for different hyperparameter choices over three ACS single-task datasets and $\epsilon = \{0.15, 1\}$. PGM parameter bin-size is varied by selecting from $\{10, 20, 50, 100\}$. RAP++ parameter for the number of epochs for linear thresholds ($T_{LT}$) is varied by selecting from $\{10, 30, 50, 70\}$.

**ML Evaluation.** We also evaluate the quality of synthetic data for training linear logistic regression models. For each dataset, we use 80 percent of the rows as a training dataset and the remainder as a test dataset. Only the training dataset is used to generate synthetic datasets subject to differential privacy. We then train a logistic regression model on the synthetic data and evaluate it on the test data. Since labels are not balanced in individual tasks, model performance is measured using F1 score which is a harmonic mean of precision and recall. Also there isn't a clear definition of the positive class in each task, so we report the macro F1 score, which is the arithmetic mean of F1 scores per class. As a "gold-standard" baseline, we also train a model on the original training set directly (i.e. without any differential privacy protections).

The ML evaluation contains both single-task and multitask experiments. Since single-task datasets only define one column as the target label for prediction, we train one logistic regression for each single-task dataset. Figure 4 summarizes our results on single-task datasets and Fig. 5 shows results on multitask ones. On single-task, RAP++ clearly outperforms every other benchmark in terms of F1 score across all tasks and privacy levels. Due to the space limit, we only show results for three tasks and on a single state, while the remaining set of experiments can be found in the appendix.

For multitask experiments, we observe that our algorithm most significantly outperforms PGM in terms of F1 score on the income task, where numerical features are more important. On the coverage tasks RAP++ wins when $\epsilon$ is small. And finally, PGM does slightly better on the employment task where numerical columns provide no information.To understand when numerical features are important, we conducted a simple experiment where we trained a logistic regression model on all features (including numerical) and compared against a model trained only on categorical features. We show that numerical features are important for the income task and less important for other tasks. See appendix for details on this. Our experiments confirm that RAP++ produces synthetic data which can train more accurate linear models when the numerical features are important for the prediction task.

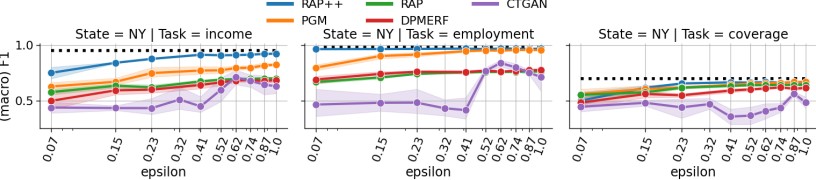

Figure 4: **ACS Single-task ML**: Comparison of synthetic data generation approaches by the F1 score achieved on linear models trained on synthetic data and F1 score achieved by training on original dataset(black dotted line). Results averaged across three single-task datasets (ACS-NY state) for different privacy levels. See Appendix for other states.

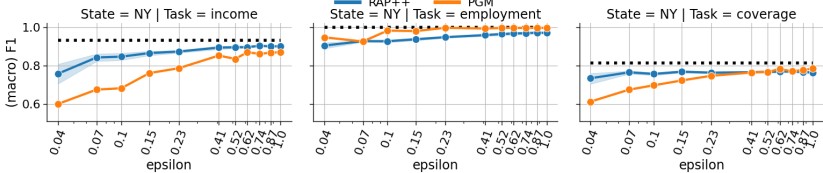

Figure 5: **ACS Multitask ML**: Comparison of synthetic data generation approaches using multitask datasets by the F1 scores achieved on linear models trained on synthetic data and F1 score achieved by training on original dataset(black dotted line). See Appendix for other states.

## 6 Limitations and Conclusions

We propose an algorithm for producing differentially private synthetic data that improves on prior work in its ability to handle numeric valued columns, handling them natively rather than binning them. We show that this leads to substantial runtime and accuracy improvements on datasets for which the numerical columns are numerous and relevant. For machine learning tasks on categorical data or for data for which the numerical features are not informative, prior methods like PGM can still sometimes produce more accurate synthetic data, although we still generally outperform in terms of runtime. Producing synthetic data useful for downstream learning beyond linear classification largely remains open: In our experiments, more complicated models trained on the synthetic data generally resulted in performance that was comparable or slightly worse than the performance of linear models trained on the synthetic data, even when the more complex models outperformed linear models on the original data. Note that it is not so much that the non-linear methods perform poorly in an absolute sense, but they fail to realize the performance gains beyond that achievable by linear models when applied to the private synthetic data. Producing high quality synthetic data that can enable downstream machine learning that obtains higher accuracy than linear models is a very interesting question.

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
