## A  Additional Related Work

Our work is most related to RAP [ABK+21], which leverages powerful differentiable optimization methods for solving a continuous relaxation of their moment matching objective. [LVW21] also leverages such differentiable optimization strategy, but their algorithm GEM optimizes over probability distributions parametrized by neural networks. Both of them focus on categorical attributes.

The theoretical study of differentially private synthetic data has a long history [BLR08, DNR+09, RR10, HT10, HR10, GRU12, NTZ13]. In particular, both RAP [ABK+21] and our work are inspired by the the theoretically (nearly) optimal *projection mechanism* of Nikolov, Talwar, and Zhang [NTZ13], which simply adds Gaussian noise to statistics of interest of the original dataset, and then finds the synthetic dataset (the "projection") that most closely matches the noisy statistics (as measured in the Euclidean norm). The projection step is computationally intractable in the worst case. Both RAP and our algorithm consider a continuous relaxation of their objective and leverage differentiable optimization methods to perform projection efficiently.

It is known to be impossible to privately learn (or produce synthetic data for) even one-dimensional interval queries over the real interval in the worst case over data distributions [BNSV15, ALMM19]. These worst case lower bounds need not be obstructions for practical methods for synthetic data generation, however. Linear threshold functions over real valued data can be privately learned (and represented in synthetic data) under a "smoothed analysis" style assumption that the data is drawn from a sufficiently anti-concentrated distribution [HRS20].

## B  Missing from Preliminaries (section 2)

For completeness, we include the definition of prefix marginal queries:

**Definition 13** (Prefix Marginal Queries). *A $k$-way prefix query is defined by a set of numerical features $R$ of cardinality $|R| = k$ and a set of real-valued thresholds $\tau = \{\tau_j \in \mathbb{R}\}_{j \in R}$ corresponding to each feature $j \in R$. Let $\mathcal{X}(R, \tau) = \{x \in \mathcal{X} : x_j \leq \tau_j \quad \forall_{j \in R}\}$ denote the set of points where each feature $j \in R$ fall bellow its corresponding threshold value $\tau_j$. The prefix query is given by*

$$q_{R,\tau}(x) = \mathbb{1}\{x \in \mathcal{X}(R, \tau)\}.$$

Model performance is measured using F1 score which is a harmonic mean of precision and recall. Also there isn't a clear definition of the positive class in each task, so we report the macro F1 score, which is the arithmetic mean of F1 scores per class.

**Definition 14** (F1 score). *Given a dataset with binary labels $D = \{(x_i, y_i)\}_{i=1}^{N}$ where $x_i \in \mathbb{R}^d$ and $y_i \in \{0, 1\}$, $\forall i \in \{1, 2, ..., N\}$, the F1-score of predictions produced by a classification model $f : \mathbb{R}^d \to \{0, 1\}$ is defined as*

$$F1(D, f) = \frac{2}{precision(f, D)^{-1} + recall(f, D)^{-1}}$$

*where*

$$precision(f, D) = \frac{\sum_{i=1}^{N} \mathbb{1}(y_i = 1, f(x_i) = 1)}{\sum_{i=1}^{N} \mathbb{1}(f(x_i) = 1)}, recall(f, D) = \frac{\sum_{i=1}^{N} \mathbb{1}(y_i = 1, f(x_i) = 1)}{\sum_{i=1}^{N} \mathbb{1}(y_i = 1)}$$

## C  Differential Privacy Analysis

Here we state the privacy theorem of RAP++ (algorithm 2). We restate definition 6 and definition 7 here:

**Definition** (Differential Privacy [DMNS06]). *A randomized algorithm $\mathcal{M} : \mathcal{X}^n \to \mathcal{R}$ satisfies $(\epsilon, \delta)$-differential privacy if for all neighboring datasets $D, D'$ and for all outcomes $S \subseteq \mathcal{R}$ we have*

$$Pr[\mathcal{M}(D) \in S] \leq e^\epsilon Pr[\mathcal{M}(D') \in S] + \delta.$$

**Definition** (Zero-Concentrated Differential Privacy [BS16b]). *A randomized algorithm $M : \mathcal{X}^n \to \mathcal{R}$ satisfies $\rho$-zero concentrated differential privacy (zCDP) if for any neighboring dataset $D, D'$ and for all $\alpha \in (1, \infty)$ we have*

$$D_\alpha(M(D) \| M(D')) \leq \rho\alpha$$

*where $D_\alpha$ is the $\alpha$-Rényi divergence.*

We use the basic composition and post processing properties of zCDP mechanisms for our privacy analysis.

**Lemma 3** (Composition [BS16b]). *Let $\mathcal{A}_1 : \mathcal{X}^n \to R_1$ be $\rho_1$-zCDP. Let $\mathcal{A}_2 : \mathcal{X}^n \times R_1 \to R_2$ be such that $\mathcal{A}_2(\cdot, r)$ is $\rho_2$-zCDP for every $r \in R_1$. Then the algorithm $\mathcal{A}(D)$ that computes $r_1 = \mathcal{A}_1(D)$, $r_2 = \mathcal{A}_2(D, r_1)$ and outputs $(r_1, r_2)$ satisfies $(\rho_1 + \rho_2)$-zCDP.*

**Lemma 4** (Post Processing [BS16b]). *Let $\mathcal{A} : \mathcal{X}^n \to R_1$ be $\rho$-zCDP, and let $f : R_1 \to R_2$ be an arbitrary randomized mapping. Then $f \circ \mathcal{A}$ is also $\rho$-zCDP.*

Together, these lemmas mean that we can construct zCDP mechanisms by modularly combining zCDP sub-routines. Finally, we can relate differential privacy with zCDP:

**Lemma 5** (Conversions [BS16b]).

1. *If $\mathcal{A}$ is $\epsilon$-differentially private, it satisfies $(\frac{1}{2}\epsilon^2)$-zCDP.*

2. *If $\mathcal{A}$ is $\rho$-zCDP, then for any $\delta > 0$, it satisfies $(\rho + 2\sqrt{\rho \log(1/\delta)}, \delta)$-differential privacy.*

We restate theorem 1 with its proof.

**Theorem** (Privacy analysis of RAP++(algorithm 2)). *For any dataset $D$, any query class $Q$, any set of parameters $K$, $T$, $\hat{n}$, $\sigma_1$, and any privacy parameters $\epsilon$, $\delta > 0$, Algorithm 2 satisfies $(\epsilon, \delta)$-differential privacy.*

*Proof.* The proof follows from the composition and post-processing properties of $\rho$-zCDP (see lemma 3 and lemma 4), together with the privacy of both the RN (definition 9) and Gaussian mechanisms (definition 8) (see lemma 2 and lemma 1 respectively).

Algorithm 2 takes as input privacy parameters $\epsilon$, $\delta$ and chooses a zCDP parameter $\rho$ such that $\epsilon = \rho + 2\sqrt{\rho \log(1/\delta)}$. Then, algorithm 2 makes $T$ calls to the RN mechanism with zCDP parameter $\rho/2T$, and makes $T \cdot K$ calls to the Gaussian mechanism with zCDP parameters $\rho/(2TK)$ to sample a sequence of statistical queries, which is then used in step 9 to generate a sequence of synthetic datasets $\widehat{D}_1, \ldots, \widehat{D}_T$. By the composition property, releasing this sequence of statistical queries satisfies $\rho$-zCDP and by post-processing the sequence of synthetic datasets satisfies $\rho$-zCDP.

Therefore, algorithm 2 satisfies $\rho$-zCDP and by the way $\rho$ was set in step 2 and the conversion lemma 5 proves the theorem.

$\square$

# D ACS Datasets

Missing details describing the datasets used in our experiments. We use datasets and tasks released by the American Community Survey (ACS) from [DHMS21]. We focus primarily on the five largest states in the U.S. Each single-task dataset defines a label column that is the target for a prediction task and a set of both categorical and numerical features. Refer to table 1 for the number of features and rows on each single-task dataset considered in our experiments. The set of multitasks datasets combines the features of all single-task datasets and contain five label columns. See table 4 for a description of all features used and for details about how each task is constructed.

| Task | Features | | Rows | | | | |
|------|----------|-----------|--------|--------|--------|--------|--------|
| | Categorical | Numerical | NY | CA | TX | FL | PA |
| income | 9 | 9 | 100513 | 183941 | 127039 | 91438 | 66540 |
| employment | 18 | 10 | 196276 | 372553 | 254883 | 192673 | 127859 |
| coverage | 19 | 10 | 74985 | 152676 | 100949 | 75715 | 48341 |
| travel | 13 | 9 | 88035 | 160265 | 111545 | 80314 | 58060 |
| mobility | 20 | 10 | 40173 | 78900 | 50962 | 32997 | 23824 |

Table 1: Single Task ACS datasets. This table describes the number of categorical and numerical features for each ACS task. The last five columns in the table, describes the number of rows of each dataset corresponding to the state.

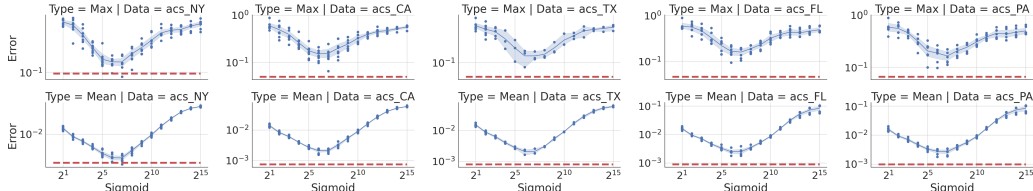

Figure 6: Comparison on five datasets of the Maximum error (left) and the Mean error (right) of mixed marginals using our temperature annealing (red line) and those using a fixed $\sigma$ parameter (blue curve). It is clear that annealing the $\sigma$ parameter strategically during optimization leads to lower error than using a fixed $\sigma$ parameter on answering marginal queries.

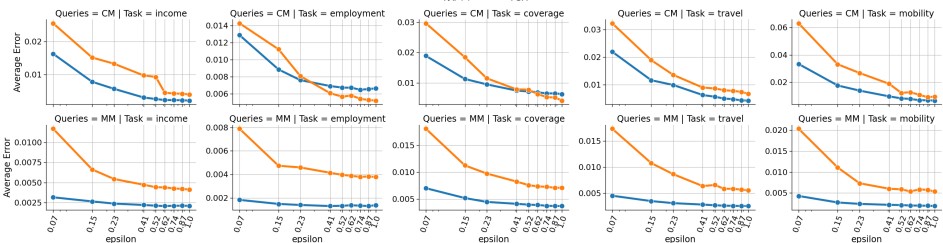

Figure 7: **Summary of average error** for Categorical and Mixed Marginals (MM) on five tasks and aggregated over five ACS states. Each column represents different task. We only compare against the most competitive approach PGM . See appendix for other approaches and CM queries. RAP++ consistently achieves lower error, specially as $\epsilon$ decreases.

## E   Implementation Details

Our algorithm is implemented in JAX [BFH$^+$18] with GPU support. DP-MERF and GAN-based algorithms are implemented in PyTorch [PGM$^+$19] with GPU support as well. PGM provides support on both CPU and GPU, however, in our experiments, we found that the algorithm ran significantly faster on CPU, so results for PGM were obtained on CPUs. To provide error bars, each algorithm is run four times with different random seeds. For fair comparison, after running several hyperparameter combinations for each algorithm, a single hyperparameter setting that works the best across various tasks and epsilon values is selected, and its results are presented in the following plots. An additional privacy budget of $\epsilon = 5$ was allocated for the preprocessing stage of CTGAN ; the reported budget for each experiment is solely used for private gradient optimization. We used the OpenDP implementation of CTGAN [GHV20] and tuned the learning rate, batch size, and noise scaling hyperparameters.

## F   Additional plots

The main body of the paper only shows results for a subset of the ACS datasets. This section we shows the remaining results for all ACS tasks and all five states. Figure 6 shows experiments for the sigmoid temperature optimization technique described in section 3 that include more states than presented in fig. 1. We also include more results for error on marginal queries. Figure 7 shows that the main advantage of RAP++ over PGM is on answering mixed marginal queries. The error over categorical marginals is comparable for both methods.

Then fig. 8 and fig. 9 we have machine learning performance on all states and all tasks. The plots show that on a large number of datasets our mechanism performs better or no much worse than all of our benchmark algorithms. See table 2 for a description of all mechanism used for comparison. In particular RAP++ over performs on the income tasks where numerical features have the most importance. To show importance of numerical features, we conduct an experiment where we train a model that ignores numerical features and compare its performance in terms of F1 score against a model trained on all features. Table 3 shows the magnitude of performance drop on each tasks when numerical features are not used to train the model. Since the income task has the largest drop we

conclude that numerical features are important for the income tasks, whereas categorical features are enough to train a linear model for other tasks.

| Algorithm | PGM | DP-MERF | DP-CTGAN | RAP | RAP++ |
|---|---|---|---|---|---|
| Citation | [MSM19] | [HAP21] | [FDK22] | [ABK$^+$21] | (This work) |
| Input Data Type | Categorical | Numerical | Mixed-Type | Categorical | Mixed-Type |

Table 2: Synthetic data mechanisms in our evaluations. Other than GAN-based approaches, most previous work only support discrete data as input, whereas RAP++ supports mixed-type data .

|  | Task F1 score | | | | |
|---|---|---|---|---|---|
| Train data Column Type | Income | Employment | Coverage | Travel time | Mobility |
| Categorical Only | 0.77 | 1 | 0.77 | 0.78 | 0.57 |
| Categorical and Numerical | 0.93 | 1 | 0.82 | 0.79 | 0.58 |

Table 3: This table compares the F1 scores for predicting five tasks on ACS NY(multi-task) of a linear model trained only on categorical features and a linear model trained on all features. It shows that for some tasks (i.e., *Employment*, *Travel time*, and *Mobility*), numerical features are not very informative, whereas for others (especially *Income*) they are crucial.

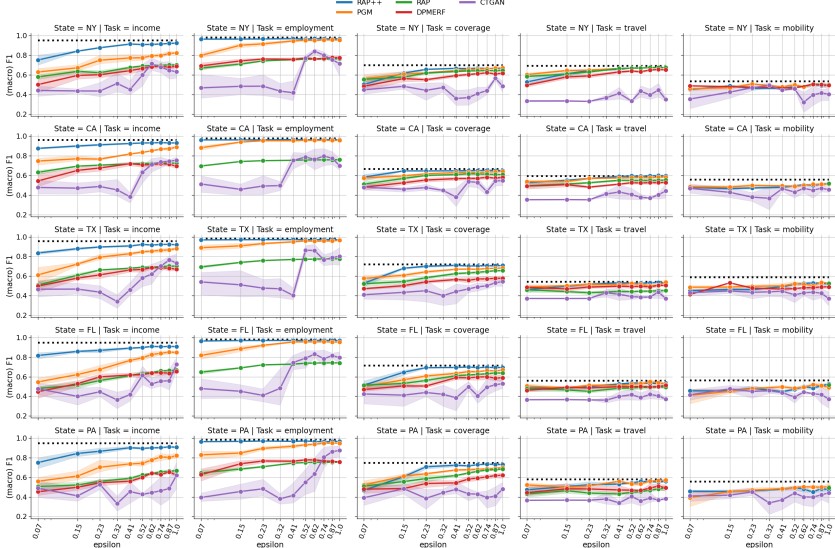

Figure 8: **ACS Single-task ML**: Comparison of synthetic data generation approaches by the F1 score achieved on linear models trained on synthetic data and F1 score achieved by training on original dataset(black dotted line). Results averaged across 25 single-task datasets for different privacy levels.

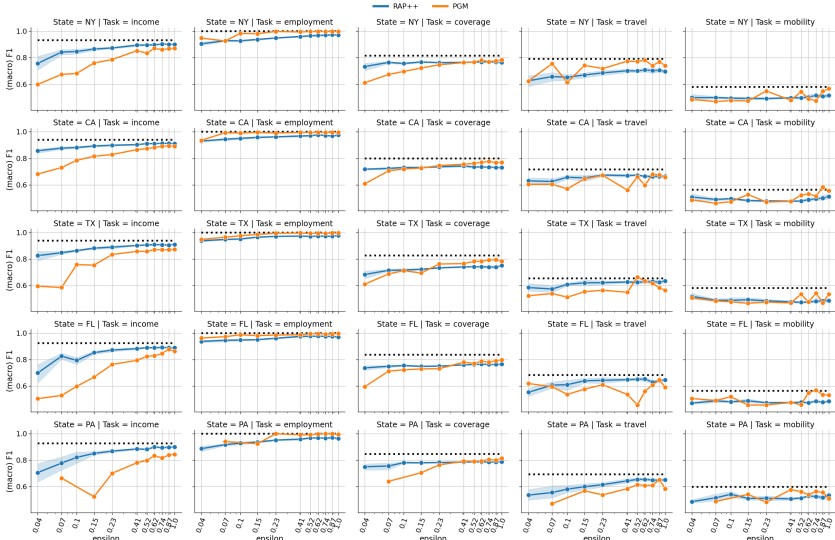

Figure 9: **ACS Multitask ML**: Comparison of synthetic data generation approaches using multitask datasets for five states by the F1 scores achieved on linear models trained on synthetic data and F1 score achieved by training on original dataset(black dotted line).

| Name | Type | Description | income | employment | coverage | travel | mobility | multitask |
|---|---|---|---|---|---|---|---|---|
| COW | CAT | Class of worker | ✓ | | | | ✓ | ✓ |
| SCHL | CAT | Educational attainment | ✓ | ✓ | ✓ | ✓ | ✓ | ✓ |
| MAR | CAT | Marital status | ✓ | ✓ | ✓ | ✓ | ✓ | ✓ |
| RELP | CAT | . | ✓ | ✓ | | ✓ | ✓ | ✓ |
| SEX | CAT | Male or Female. | ✓ | ✓ | ✓ | ✓ | ✓ | ✓ |
| RAC1P | CAT | Race | ✓ | ✓ | ✓ | ✓ | ✓ | ✓ |
| WAOB | CAT | World area of birth | ✓ | ✓ | ✓ | ✓ | ✓ | ✓ |
| FOCCP | CAT | Occupation | ✓ | ✓ | ✓ | ✓ | ✓ | ✓ |
| DIS | CAT | Disability | | ✓ | ✓ | ✓ | ✓ | ✓ |
| ESP | CAT | Employment status of parents | | ✓ | ✓ | ✓ | ✓ | ✓ |
| CIT | CAT | Citizenship status | | ✓ | ✓ | ✓ | ✓ | ✓ |
| JWTR | CAT | Means of transportation to work | | | | ✓ | ✓ | ✓ |
| MIL | CAT | Served September 2001 or later | | ✓ | ✓ | | ✓ | ✓ |
| ANC | CAT | Ancestry | | ✓ | ✓ | | ✓ | ✓ |
| NATIVITY | CAT | Nativity | | ✓ | ✓ | | ✓ | ✓ |
| DEAR | CAT | Hearing difficulty | | ✓ | ✓ | | ✓ | ✓ |
| DEYE | CAT | Vision difficulty | | ✓ | ✓ | | ✓ | ✓ |
| DREM | CAT | Cognitive difficulty | | ✓ | ✓ | | ✓ | ✓ |
| GCL | CAT | Grandparents living with grandchildren | | | | | ✓ | ✓ |
| FER | CAT | Gave birth to child within the past 12 months | | | ✓ | | ✓ | ✓ |
| WKHP | NUM | Usual hours worked per week past 12 months | ✓ | ✓ | ✓ | ✓ | ✓ | ✓ |
| PINCP | NUM | Total person's income | | ✓ | ✓ | ✓ | ✓ | |
| AGEP | NUM | Age of each person | ✓ | ✓ | ✓ | ✓ | ✓ | ✓ |
| PWGTP | NUM | Person weight | ✓ | ✓ | ✓ | ✓ | ✓ | ✓ |
| INTP | NUM | Interest, dividends, and net rental income past 12 months. | ✓ | ✓ | ✓ | ✓ | ✓ | ✓ |
| JWRIP | NUM | Vehicle occupancy | ✓ | ✓ | ✓ | ✓ | ✓ | ✓ |
| SEMP | NUM | Self-employment income past 12 months | ✓ | ✓ | ✓ | ✓ | ✓ | ✓ |
| WAGP | NUM | Wages or salary income past 12 months | ✓ | ✓ | ✓ | ✓ | ✓ | ✓ |
| POVPIP | NUM | Income-to-poverty ratio | ✓ | ✓ | ✓ | ✓ | ✓ | ✓ |
| JWMNP | NUM | Travel time to work | ✓ | ✓ | ✓ | | ✓ | |
| JWMNP(binary) | CAT | Commute > 20 minutes | | | | ✓ | | ✓ |
| PINCP(binary) | CAT | Income > $50K | ✓ | | | | | ✓ |
| ESR | CAT | Employment status | ✓ | | ✓ | ✓ | | ✓ |
| MIG | CAT | Mobility status (lived here 1 year ago) | | ✓ | ✓ | ✓ | ✓ | ✓ |
| PUBCOV | CAT | Public health coverage | | | ✓ | | | ✓ |

Table 4: Features in both single-task and multitask datasets. Each rows shows the name, type (numeric or categorical) and description of each ACS feature. The check mark indicates whether a features is included for a tasks. Note that all tasks included as many numeric features as possible and the multitask datasets included all features and all five labels. Features description can be found here: ACS PUMS Data Dictionary.