# OpenReview forum: "Private Synthetic Data for Multitask Learning and Marginal Queries"
_NeurIPS.cc/2022/Conference — NeurIPS 2022 Accept_

### Official Review · Reviewer_DNy5 · 2022-07-06

**Rating:** 6
**Confidence:** 4
**Soundness:** 3 good
**Presentation:** 3 good
**Contribution:** 2 fair

**Summary:**

The paper iterates upon the "Relaxed Adaptive Projection (RAP)" framework from [ABK+21], a "moment matching approach" for generating private synthetic data. The primary improvement wrt to the prior work is the ability of handling a mixture of both categorical and numerical features --- where RAP requires a discretization of the numeric domain via binning. This is achieved in two steps: (1) numeric based queries are introduced (which are different types of threshold queries); and (2) a differentiable approximation to these numeric queries are introduced with via a tempered sigmoid annealing query. The later is key in the optimization of the private synthetic data. With these tools to handle numerical based queries, differential privacy mechanisms are used to ensure that the synthetic data is private.

In summary, the approach can be broken down as follows:
1. The K worst query functions are selected wrt error and using "Report Noisy Top-K", a DP mechanism;
2. The values of these queries are calculated with the Gaussian DP mechanism;
3. The query functions in (1) are converted to their differentiable approximations;
4. A projection step occurs using results of (2) and (3) to find the best data wrt error.

These steps done in the paper's Algorithm 2. The proof of $ (\epsilon, \delta) $-DP follows from standard composition and post-processing properties of zero concentration DP.

Experimentally, the paper uses their approach to generate synthetic data for multitask learning (multiple columns of labels to target in classification). With this in mind, the paper evaluates their approach over many variants of the ACS dataset. The results present appear promising for both mixed-marginal query evaluation and (linear) classification.


**Questions:**

Comments / Suggestions / Questions
1. Comparison of Fig 2 & 5 for all approaches is not present in Appendix. This seems to be key in evaluating RAP++, especially examining the difference between RAP and RAP++ for CM to see if RAP++ is strictly better.
2. Fig 1. seems a bit odd without it specifying the experiment it comes from.
3. What if any of the bound on query error for RAP [ABK+21] can be "inherited" for RAP++? In particular for the numerical queries.
4. The initialization of synthetic dataset doesn't seem to be discussed anywhere, Algorithm 2, Line 3.
5. Algorithm 1/2 is slightly confusing as it appears to suggest that temperature scaling is utilized even for CM queries (which from digging into the code is not the case). Clarification on the specifics of the loss functions for different queries would be helpful. Furthermore, stating the case in which RAP++ reduces to RAP explicitly would be helpful for the reader (which I believe occurs when all queries are CM type).
6. The limitation section states that "Producing synthetic data useful for downstream learning beyond linear classification largely remains open". Do non-linear classifiers perform poorly on RAP++? Plots of failure cases would be interesting to see for this case.

Minor typos
  - Algorithm 1. Line 5: "Stating" -> "Starting"
  - Algorithm 1. Line 5: Should "$D_{k}$" be "$D_{j}$"?
  - Algorithm 1. Line 5: Should "[...] descent on $\hat{D}$" be "[...] descent on $L_{j}(\hat{D})$"?
  - Line 227: "Eq. 2" is inconsistent with equation citation in previous parts of the paper.

**Limitations:**

The author states the limitation for ML downstream tasks to that of only linear classification. It would be useful to see explicit experiments on non-linear classifiers trained on RAP++ synthetic datasets.

---

Edit: Addressed in the reviewer discussion.

**Strengths And Weaknesses:**

Strengths
  - The paper presents an iterative improvement over RAP. This is particularly shown in the experimental section, where RAP++ is competitive or superior to other prior work.
  -  The explanation and motivation for the tempered sigmoid, the subsequent queries, and annealing approach was good.

Weaknesses
  - Although there is a privacy guarantee and promising experimental results, there is not an theoretical error analysis. From briefly looking at [ABK+21], RAP has a bound on query error.
  - There are a few unclear aspects of the paper.
  - Fig 2 & 5 cites the Appendix for comparison against other approaches, but this does not appear in the Appendix.

Points which could be rephrased as questions / comments are reiterated below.

---

> ### Author Response · Authors · 2022-08-02
> **Response to Reviewer DNy5**
>
> Thank your for your review! Here are our answers to your questions:
>
> **RAP vs. RAP++ on Categorical Marginal queries:** In most experiments, we used MWEM+PGM as our primary baseline because it was the most performant of the various approaches we tried (MWEM+PGM, DPMERF, CTGAN, and RAP). The difference between RAP and RAP++ is in how it treats numeric valued features and threshold based queries. In particular, for categorical marginal queries, RAP and RAP++ are essentially the same algorithm, and so it does not make sense to compare RAP and RAP++ in this setting. We'll update the caption to our figure and add discussion to reflect this.
>
> **Figure 1:** We intended figure 1 to communicate a qualitative sense for how our annealing approach helps, but we agree that it might be confusing at the point that it comes in the paper --- we will add exposition to describe more details of the experiment it comes from (that at the moment would only make sense to a reader who had read further into the paper, since we have not yet described out datasets at the point that figure 1 appears). Thanks for pointing this out!
>
> **Accuracy guarantees from RAP:** The accuracy analysis for RAP assumes that the optimization step is solved perfectly, and then only uses the fact that the queries to be answered are of bounded cardinality, and that the queries themselves are Lipschitz continuous. As such, our method, which is an instantiation of RAP with new query classes, inherits RAP's accuracy guarantees (again, under the assumption that the optimization problem can be solved). Threshold queries are not Lipschitz continuous (indeed, all threshold queries cannot be answered accurately subject to differential privacy in the worst case because they have unbounded Littlestone dimension --- this is a result of [ALMM19] which we cite). But our Sigmoid approximations to them are Lipschitz, with Lipschitz parameters determined by the choice of sigmoid temperature. We will add a discussion of RAP's accuracy guarantees and the sense in which we inherit them to the paper.
>
> **Initialization of the synthetic dataset:** We always initialize the synthetic dataset uniformly at random over the dataset domain. We will update our pseudocode to specify this --- thanks for catching the oversight!
>
>
> **Temperature scaling and CM queries:** You are right that the temperature scaling is only relevant to MM queries and that for workloads consisting only of CM queries, RAP++ reduces to RAP. We will clarify this in our algorithm and discussion.
>
> **Beyond Linear Models:** In our experiments, more complicated models trained on the synthetic data generally resulted in performance that was comparable or slightly worse than the performance of linear models trained on the synthetic data, even when the more complex models outperformed linear models on the original data. (So its not so much that they perform poorly, but they fail to realize performance gains beyond that achievable by linear models). This is why we say that producing synthetic data for downstream tasks beyond linear classification remains open. We will add detail and elaborate on our preliminary (non)findings in the non-linear case.

---

> > ### Comment · Reviewer_DNy5 · 2022-08-08
> > **Thank you for the response**
> >
> > Thanks you for the response!
> > I am happy with the responses given. I think that the clarification given in your response regarding the relation of RAP and RAP++ will help readers not entrenched in the literature have a clear picture of the lineage of your approach. I also believe that the discussion on accuracy in Reviewer vqTp's response and the response above would also strengthen the paper.
> >
> > I'll lightly edit my review above to indicate that the limitation I listed has been addressed.
> > I won't change the review numerically, but will do so shortly after discussion with other reviewers.

---

> > > ### Author Response · Authors · 2022-08-08
> > > **Thanks**
> > >
> > > Terrific --- and thank you for the time you spent reviewing our paper! Your feedback has been valuable.

---

### Official Review · Reviewer_vqTp · 2022-07-19

**Rating:** 6
**Confidence:** 3
**Soundness:** 3 good
**Presentation:** 3 good
**Contribution:** 3 good

**Summary:**

This paper provides a method for generating synthetic differentially-private datasets for use in answering statistical queries without the need for binning.  Specific query types that were analyzed include Mixed Marginal Queries, Class Conditional Linear Threshold Queries, and "Querying the Error."  The is an improvement over previous work for certain classes of statistical queries.

**Questions:**

Can you address what bounds are given for this algorithm, eg on added error?
Where do the main technical innovations of this paper lie?

**Limitations:**

Yes, the limitations have been addressed.

**Strengths And Weaknesses:**

The main improvement in this paper is for the case of mixed-type queries queries, which contain a mixture of categorical and numerical features. The technique in the paper is an extension of the "relaxed projection mechanism" by considering the k hardest queries and also adding a simulated annealing step.  The experimental results are also impressive.  Overall, this would be a worthwhile contribution.

However, I do have some concerns: One question I had is about the lack of theorems in the paper.  Epsilon-delta privacy follows straightforwardly from the addition of the noise, how about other components, eg added error?  Another thing that would help if the authors explained the where the main difficulty of applying these known techniques lies.

---

> ### Author Response · Authors · 2022-08-02
> **Response to Reviewer vqTp**
>
> Thanks for your review!
>
> **Accuracy Guarantees:**
> You are right that our algorithm does not come with a worst-case accuracy guarantee, and this is for fundamental reasons --- in general, no computationally efficient algorithm with provable privacy guarantees can have provable accuracy guarantees that exceed those of the simple Gaussian mechanism. This is a result of Ullman '16: (Ullman, Jonathan. "Answering $n^2+o(1)$ counting queries with differential privacy is hard." SIAM Journal on Computing 45.2 (2016)) which we can cite and discuss in the paper. Because of this fundamental barrier, all computationally efficient algorithms for generating synthetic data that have provable privacy guarantees must have heuristic accuracy guarantees. There are some ways to partially circumvent this: for example, RAP has accuracy guarantees for the continuous valued queries it answers under the assumption that the optimization is in fact able to solve the ``projection'' problem. These were proven in [ABK+21] and [NTZ13]. Our algorithm is an application of RAP (with a new query class and optimization procedure), and so inherits these same accuracy guarantees (once again, conditional on the success of the optimization). We can add a discussion of this in the paper as well. The main novelty in our paper (and, we emphasize again, necessarily -any- paper giving provably private and computationally efficient algorithms for synthetic data generation) is therefore solving a number of design challenges that allow us to get provable privacy guarantees (using standard analyses) with new, state of the art empirical accuracy findings.
>
> **Technical Innovation:** The primary technical innovation in our paper is in our use of Sigmoid approximations to threshold functions, which allows us to apply continuous optimization machinery. Although this is a natural idea, for reasons that we discuss in the paper, it does not work well on its own, because for any setting of the sigmoid parameter, either the differentiable queries described via the sigmoid approximation are poor approximations of the original query, or they are hard to optimize. We are able to make this idea work through our dynamic scaling of the sigmoid parameter (which we call the sigmoid temperature). This idea, of dynamically changing the queries we are optimizing for over the course of the optimization, has not appeared before and turned out to be the key to making our approach work. Another benefit of this approach is that we do not have to tune the sigmoid parameter of our approximations, eliminating what would otherwise be a  hyper-parameter.

---

### Official Review · Reviewer_cmFo · 2022-07-21

**Rating:** 6
**Confidence:** 3
**Soundness:** 3 good
**Presentation:** 4 excellent
**Contribution:** 3 good

**Summary:**

The paper considers the problem of privately generating synthetic datasets based on a private dataset for which each data point is a mix of numerical and categorical variables. The best-performing past approach in this area is an approach called moment-matching, which tries to generate a synthetic dataset that minimizes the error of marginal queries (i.e., queries that take 0-1 variables x_i, and ask for the expectation of $\prod_{i \in S} x_i$) made on the synthetic database. They build off an algorithm called RAP, which repeatedly (privately) chooses new marginal queries for which the current synthetic database has high error, and then tries to minimize the error on all queries chosen so far. Past approaches made numerical variables suitable for marginal queries by placing them into bins (thus making them categorical); the authors argue the binning heuristic is impractical. They instead propose two new classes of queries. The first applies a threshold to the numerical variables to turn them into 0-1 variables. The second applies a linear classifier to a subset of the numerical variables to generate a 0-1 variable.

The most challenging step in the RAP algorithm is to, after choosing a set of queries, find a synthetic dataset that minimizes the error on these queries. RAP does this by defining a fractional relaxation of synthetic datasets and an error function over fractional datasets, which allows one to use continuous optimization techniques. The authors propose RAP++, which is RAP, but also with the option to choose queries from their two proposed classes. To make the error function differentiable, the authors approximate the threshold/linear classifier in the queries with a sigmoid function. However, a sigmoid function has to either be highly non-smooth (which makes it hard to optimize over) or poorly approximate threshold functions. The authors propose that rather than doing a one-shot optimization over the error function defined using sigmoids, instead using an annealing-style algorithm that repeatedly solves the optimization problem, but each time decreases the temperature of the sigmoids.

In experiments, the authors compare RAP++ to several other synthetic data generation methods in the literature. The authors show that RAP++ produces synthetic datasets with better accuracy/error than the other methods in most settings. However, since RAP++ places less emphasis on marginal queries just on categorical data in its training process, in tasks where numerical data is not useful it is outperformed by some other methods. The authors also show the runtime of RAP++ is much less than the runtime of the next-best method in their experiments.

**Questions:**

My two main questions/concerns are regarding the experiments:

-How were the algorithms the authors compared against chosen? In particular, it seems like in [TMH+21] https://arxiv.org/pdf/2112.09238.pdf (which the authors cite), they state that an algorithm called MST frequently performs well in a series of benchmarks; in particular, it seems to outperform 3 of the 4 benchmark algorithms tried in this paper. However, the authors did not compare to MST. Was there a reason MST wasn't chosen?

-As far as I can tell, even after looking in the appendix, the only benchmark to which the authors made a runtime comparison is PGN. This seems to be motivated by the fact that PGN is the best performing of the benchmarks. However, the results presented in the paper don't seem to preclude the possibility of another algorithm having better runtime/scalability than RAP++ at the cost of having a higher error as well.

EDIT: After reading the authors' rebuttal, I feel both questions have been addressed and are not of concern.

**Limitations:**

Aside from the concerns in "questions", I don't believe there are any major limitations or negative social impacts not addressed by the authors.

**Strengths And Weaknesses:**

Originality: To the best of my knowledge, while the algorithm in this paper builds upon a pre-existing algorithm (RAP), the two improvements made to that algorithm (new types of queries, and an annealing method for optimization) are both fairly original. Furthermore, the problem is fairly well-studied, so a new approach for the problem is more novel than perhaps, say, a new approach appearing in the second or third paper on a problem.

Quality: The methods the author propose appear to be very natural despite their originality/novelty. The privacy guarantees of the algorithm are sound and easy to see. The experiments appear to be fairly extensive and support the authors' claims. I do have some questions/concerns about the experiments, however, which I have placed in the questions section of the review.

Clarity: I thought that the paper was very clean from an exposition standpoint. The introduction does a good job explaining the issue with past approaches, and when explaining the algorithm I felt the design decisions the authors made were well-motivated by their explanation of the problems those decisions addressed. There are some clarity issues with the experimental design/results related to my questions/concerns from the previous bullet.

Significance: I think the paper is fairly impactful; synthetic data generation is a well-studied problem that many practitioners are trying to solve in production. In turn, algorithms like the RAP++ algorithm proposed in this paper have a reasonable chance to be deployed in practice on wide scales.

---

> ### Author Response · Authors · 2022-08-02
> **Response to Reviewer cmFo**
>
> Thank you for your careful review!
>
> **Choice of Comparison Algorithms:** We chose MWEM+PGM as the exemplar algorithm to compare to for the category of "moment matching" methods. Many competitive methods are variants of PGM (including MST).  McKenna et al. (https://arxiv.org/pdf/2201.12677.pdf) summarizes in Table 1 properties of different instantiations of PGM.  We selected MWEM+PGM because it is memory efficient for high-dimensional datasets (is "workload aware"). MST is not memory efficient and could not be run on the datasets used in our experiments. Other variants of PGM (e.g. AIM) differ via heuristics that are added in a modular fashion and could be applied also to other algorithms like RAP. For example, AIM differs from MWEM+PGM in that it uses a dynamic strategy to allocate privacy budget at each round. We think of this as an optimization (that may or may not help for different tasks) that can be added on to any iterative algorithm, and so to perform the most direct head to head comparison we use the same simple (non-adaptive) budget allocation scheme across all of the iterative algorithms.
>
> **Runtime Comparisons:** We compare RAP++ to MWEM+PGM in runtime because MWEM+PGM was the most competitive of our comparison algorithms in terms of accuracy. Amongst the algorithms we compare to, DP-MERF runs more quickly, but its accuracy is not competitive, and so we did not do a detailed run-time comparison. We can add a discussion of this to the paper.

---

> > ### Comment · Reviewer_cmFo · 2022-08-07
> > **Thank you for the response**
> >
> > Thank you for the detailed response! After reading I believe my concerns were are well-addressed by the response. I will read the discussions with the other reviewers and then reconsider my review.

---

> > > ### Author Response · Authors · 2022-08-08
> > > **Thanks**
> > >
> > > Great --- and thank you again for the time you spent reviewing our paper, which has been valuable to us.

---

### Meta-Review · Area_Chair_Gdk7 · 2022-08-30

**Recommendation:** Accept
**Confidence:** Less certain

**Metareview:**

This paper provides a method for generating synthetic differentially-private datasets for use in answering statistical queries, including Mixed Marginal Queries, Class Conditional Linear Threshold Queries, and "Querying the Error." The is an improvement over previous work.  A solid paper that all reviewers are positive about.

**Award:**

No

---

### Decision · Program_Chairs · 2022-09-14

Accept